

# *The Determination of Surfactants at the Sea Surface*

Leon King[1], Ieuan J. Roberts[1], Liselotte Tinel[1], Lucy J. Carpenter[1]

[1]Wolfson Atmospheric Chemistry Laboratories, Department of Chemistry, University of York, York, YO10 5DQ, United Kingdom

*Correspondence to*: Lucy J. Carpenter (lucy.carpenter@york.ac.uk)

**Abstract.** The surface of the ocean is a critical yet little understood interface that covers more than 70% of the Earth's surface. Evidence is emerging that the so-called sea surface microlayer (SML) - the thin film of the ocean surface which is enriched in surface active material and contains large chemical, physical and biological gradients that separate it from the underlying seawater - plays an important role in regulating the air-sea exchange of gases and aerosols. Indeed, recent studies have

suggested that (a) there is a ubiquitous enrichment of surfactants in the SML even at high wind speeds; (b) surfactants exert a control on air-sea $CO_2$ exchange at the ocean basin scale, even at high wind, high latitude oceans, and (c) interfacial photochemistry within the SML serves as a major global abiotic source of volatile organic compounds (VOCs), competitive with emissions from marine biology. These conclusions are based on measurements of "surfactant activity" (SA) from alternating current (AC) voltammetry, showing enrichment of SA in the SML compared to subsurface waters at the ocean

basin scale even at high wind speeds, and a relationship between SA and suppression of air-sea gas exchange. SA is calibrated using the large non-ionic surfactant Triton X-100 (TX-100) and expressed in concentration units of TX-100 equivalents. Here, we show that the response of SA-voltammetry varies widely for different surfactants, depending on the surfactant's molecular weight and its charge. Further, even at short deposition times of 15 s, the response becomes saturated above total surfactant concentrations of 1-2 mg L$^{-1}$, which are at the high end of those observed in the SML. This behaviour was also observed when

comparing measurements of seawater and lake water by SA voltammetry to surface film pressure ($\Delta\gamma$) measured by tensiometry. These two different methods for assessing the presence of surfactants showed that, while SA generally increases as surface film pressure increases, the correlation is poor and SA values plateau above ~2 mg L$^{-1}$ TX-100 eq. The implications of these results are that SA might not accurately capture variations in soluble and insoluble surfactants present in ocean waters.

## 1 Introduction

### 1.1 Composition and structure of the SML

The sea-surface microlayer (SML) is the thin boundary layer forming the interface between the atmosphere and the oceanic mixed layer. It comprises the top 10-1000 µm of the ocean and has different physicochemical properties, biological properties and composition compared to the underlying ocean (Liss and Duce, 1997). The SML consists of several sublayers; the viscous sublayer (>1000 µm), the thermal sublayer (500 µm) and the diffusion sublayer (50 µm) (Soloviev and Lukas, 2014) (note that





these depths are not fixed but will change depending upon physical stresses, Salter, 2010), as well as potential surfactant films

formed by either anthropogenic or natural sources (Soloviev and Lukas, 2014). A schematic of processes in the SML is shown

in Fig. 1.

**Insert Figure 1 here**

Organics in the SML are not distributed evenly in either latitude/longitude or in depth, due to their multiple sources and

hydrophobicities. Estep et al. (1985) suggested that particulate matter would adhere to the surface monolayer due to the effects

of surface tension but dissolved organic matter (DOM) would not be concentrated in the monolayer and instead form a

graduated structure based on hydrophobicity, and possibly influenced by other properties, throughout the rest of the SML.

Underneath this monolayer exists a graduated layer of variable thickness containing a range of classes of DOM including

amino acids (Kuznetsova and Lee, 2001), carbohydrates (Compiano et al., 1993) and proteins.


When surface active substances, or surfactants, are present in the SML in high enough concentrations they will form a

monomolecular film of about 2-3 nm thickness often referred to as a 'slick' (Gade et al., 2006). The presence of surfactants at

the surface causes a decrease in surface tension due to the surfactant molecules disrupting the strong hydrogen bonds between

water molecules. Surfactants in the open ocean are formed of a wide variety of compounds including lipids, polysaccharides

and proteins (Sakugawa et al., 1985; Sakugawa and Handa, 1985), chromophoric DOM (Tilstone et al., 2010) and transparent

exopolymer particles (Wurl and Holmes, 2008). The surfactants in the ocean are primarily formed by bacteria (Kurata et al.,

2016), phytoplankton (Žutić et al., 1981) and zooplankton grazing (Kujawinski et al., 2002), with small quantities coming

from terrestrial sources (Donelan et al., 2002) and photochemical reactions of organics (Tilstone et al., 2010). Surfactant

accumulation in the SML occurs partially through bubble scavenging from the underlying waters (Tseng et al., 1992), where

bubbles rising through the water column accumulate surfactants and other compounds that are then added to the SML when

they burst at the surface. This scavenging also aids the rapid reformation of the SML when it is disturbed by breaking waves,

since the bubbles formed by the waves collect the compounds and return them to the surface (Tseng et al., 1992).

**1.2 Surface tension, surfactant activity (SA), and their measurement in seawater**

Surface tension ($\gamma$) and surface film coverage ($\Delta\gamma$) are commonly measured properties of monolayers (Schmidt and Schneider,

2011). Surface tension is a thermodynamic property related to the cohesive forces between liquid molecules and has units of

force per unit distance. Water has a high surface tension (72.8 mN m$^{-1}$ at 20 °C) due to the high attraction of water molecules

to each other. The surface tension for mixed systems, like seawater, is determined by the surface excess of each component,

$\Gamma$, and their individual chemical potentials ($\mu$):

$$-d\gamma = \Gamma_1 d\mu_1 + \Gamma_2 d\mu_2 \,, \tag{1}$$

The surface excess concentration, $\Gamma$, is the amount of solute present per unit area of surface, in excess of the amount that would

be present if the bulk concentration were uniform right up to the surface. A positive surface excess (surfactants) will cause a





reduction in the overall surface tension, whereas a negative surface excess (inorganic salts) will cause an increase in the surface tension.

As pointed out by Nayar et al. (2014), there have been only three major studies of the surface tension of seawater in the last 100 years, by Krummel in 1900, Chen et al. in 1994 and Schmidt and Schneider in 2011. Of these, only Schmidt and Schneider (2011) took steps to consider organic surface-active material, by using solid phase extraction to obtain surfactant-free seawater, and derived an equation that describes $\gamma$ as a function of temperature ($T$/°C) and salinity ($S$). Nayar et al. (2014), using the ASTM D1141 standard for substitute ocean water (ASTM Standard, 2008), extended the seawater measurements of surface tension beyond a temperature of 35 °C and salinity of 35 g/kg and derived a reference correlation using their own and literature data, which we use here:

$$\gamma_0 = \gamma_w [1 + 3.766 \times 10^{-4} S + 2.347 \times 10^{-6} ST] \,, \tag{2}$$

where, $\gamma_0$ is the surface tension of surfactant-free seawater in mN m$^{-1}$, $S$ is reference salinity in g/kg, $T$ is temperature in Celsius and $\gamma_w$ is the surface tension of pure water in mN m$^{-1}$.

Surface film coverage is defined by the difference between $\gamma_0$ and the surface tension of the sample (Schmidt and Schneider, 2011):

$$\Delta\gamma = \gamma_0 - \gamma \,, \tag{3}$$

Schmidt and Schneider (2011) analysed natural seawater from a coastal station in the southern Baltic Sea, sampled approximately 20–30 cm below the water surface using glass bottles, and found depressions in surface tension ($\Delta\gamma$) of up to 10 mN m$^{-1}$.

Given the importance of surface tension in governing the properties of seawater for air-sea exchange of gases and particles, it is at face value surprising there are very few measurements. This no doubt reflects the fact that tensiometry methods, whether *via* force or optical tensiometers, are not practical at sea; so far measurements have only been reported from samples brought back to the laboratory. In this study, we report some limited observations of $\gamma$ and $\Delta\gamma$ in the SML and underlying waters, adding to the very small data base of such measurements.

Sinusoidal alternating current (AC) voltammetry is a very sensitive technique for the determination of surface-active substances in natural waters, pioneered by Ćosović and co-workers (Ćosović, 1990; Ćosović and Vojvodić, 1982). The method applies a constant potential between a hanging mercury drop electrode (HMDE) and a reference electrode. This produces an electrical double layer at the mercury–water interface to which surface active substances in the water sample are adsorbed, primarily as a result of hydrophobic expulsion, as the sample is stirred for some deposition period. The change that this





adsorption causes in the 'capacitive' or 'charging' of the electrical double layer current ($\Delta I$) is related to the extent of adsorption.


A calibration curve is constructed by plotting $\Delta I$ against the concentration of a known surfactant (Ćosović and Vojvodić, 1982). Typically, the reference surfactant Triton X-100 (TX-100) is used for calibration and so-called "surfactant activity" (SA) values are commonly given in units of TX-100 equivalents. TX-100 is a large non-ionic surfactant with a hydrophilic polyethylene oxide chain and a hydrophobic alkyl substituted phenyl headgroup, chosen as a proxy for a generic surfactant

due to its solubility, stability and commercial availability (Ćosović and Vojvodić, 1998).

The AC voltammetry method has been employed by several groups to measure SA in seawater (Ćosović and Vojvodić, 1998; Gašparović and Ćosović, 2001; Sabbaghzadeh et al., 2017; Wurl et al., 2011). In the open Atlantic Ocean, Sabbaghzadeh et al. (2017) found SA values of between 0.12 and 1.77 mg L$^{-1}$ TX-100 eq. in the SML and enrichment factors (EFs: defined as

SA$_{SML}$/SA$_{SSW}$ where SSW is the sub-surface water) of predominantly > 1 and up to 4.5 in two separate cruises between 50$^{\circ}$N and 50$^{\circ}$S. Wurl et al. (2011) sampled a multitude of subtropical, temperate and polar waters and observed a range of SA concentrations in the SML of between 0.10 – 5.0 mg L$^{-1}$ TX-100 eq. Wurl et al. (2011) found EF >1 consistently at wind speeds above 5 m s$^{-1}$, with values up to 5.6, and some depletion (i.e. EF <1) observed at lower wind speeds. On this evidence, these authors proposed that surfactant enrichment of the SML could be a ubiquitous feature of the open ocean even at high

wind speeds.

These reported SA concentrations and EFs apply to the specific sampling method, and therefore thickness, of the sampled SML, as well as potentially the depth of the sampled SSW. There are a wide variety of different methods for sampling the SML. The most commonly used is the mesh or Garrett screen, which typically retrieves a SML thickness of 100 - 400 µm.

Other methods include glass plates ($\leq$ 100 µm) and membrane filters ($\leq$ 40 µm) (Cunliffe et al., 2013). As these sampling methods do not sample the same fraction of the SML, direct comparisons between samples taken by two different sampling methods is not possible. Therefore, it is not necessarily valid to compare EFs between different studies (note that Sabbaghzadeh et al. (2017) used a Garrett screen and Wurl et al. (2011) a glass plate) nor to treat EF as a definitive quantity.

It is important to note that, thus far, there appears to have been no investigation of how SA is related to surface tension or surface film pressure.

## 1.3 Potential impact of surfactants

Brüggemann et al. (2018) showed that if the SML is assumed to exist up to a wind speed limit of 13 m s$^{-1}$, as found from the SA measurements of Sabbaghzadeh et al. (2017), nearly the entire ocean surface is covered in an SML. The SML can affect

the exchange of gases between the ocean and the atmosphere through multiple processes.



Surfactant films at the air-sea interface provide an additional diffusion barrier and reduce turbulent diffusion by supressing wave formation (McKenna and Bock, 2006), thus reducing the rate of air–sea gas exchange through turbulent processes such as bubble bursting (Liss and Duce, 1997). The air-sea transfer of $CO_2$ and other climate gases can be suppressed by up to 50

% by surfactants (Nightingale, 2013; Salter et al., 2011). Many studies have found a significant dampening of the air-sea gas exchange velocity ($k_w$) by surfactants (Broecker et al., 1978; Goldman et al., 1988; Frew et al., 2002, Schmidt and Schneider, 2011; Pereira et al., 2016). A distinct decrease in $k_w$ has been observed as surface tension is decreased due to the presence of surfactants, but this depression of $k_w$ has been found to plateau above $\Delta\gamma$ of about 1 mN m$^{-1}$ (Schmidt and Schneider, 2011 and references therein). Pereira et al. (2016) on the other hand observed a linear relationship between $k_w$ suppression and SA.


The ocean surface is heavily exposed to solar radiation, resulting in photochemical processes readily occurring in the SML. Dissolved organic material (DOM) can be photodegraded by UV radiation and produce smaller volatile organic compounds (VOCs) such as carbonyls which can then be used by microorganisms in the SML, aiding biological activity (Cunliffe et al., 2011). The VOCs produced by these photochemical reactions (Brüggemann et al., 2018) can form secondary organic aerosol

(SOA) upon being oxidised in the atmosphere. SOA affects the radiative properties and lifetime of clouds, having global climate impacts (Shrivastava et al., 2017). This abiotic source of VOCs has been suggested to be of similar importance to biogenic emissions (Brüggemann et al., 2018), based on the ubiquitous presence of surfactants as determined from measurements of SA (Sabbaghzadeh et al., 2017).

Less volatile components of the SML such as surfactants and bacteria are also emitted from the ocean as part of sea-spray aerosol produced by bubbles bursting at the surface. The result of this process is an internally mixed primary marine aerosol composed of sea salt and organics, in which the organic fraction can be a major component (Oppo et al., 1999). Marine bioaerosol is associated with negative health impacts on vegetation (Bussotti et al., 1995), animals and humans (WHO, 2003). These types of aerosol are known to be particularly effective ice nuclei (Wilson et al., 2015), which cause water droplets in

clouds to freeze at higher temperatures than normal, reducing the cloud lifetime and so potentially having a warming climate effect by reducing the amount of incoming solar radiation reflected by clouds (Murray et al., 2012).

The solubility of compounds in the ocean is affected by surfactants, which influences the deposition of pollutants to the SML (Laha et al., 2009). Pollutants such as polycyclic aromatic hydrocarbons (PAHs) become more soluble in the ocean if

surfactants are present, leading to higher toxicity in the marine environment (Cincinelli et al., 2001).

In conclusion, studies using voltammetry to determine SA have concluded that the SML is a near permanent aspect of the ocean which must be considered when trying to understand and model processes at the air-sea interface (Sabbaghzadeh et al., 2017; Wurl and Holmes, 2008). The ability of the SML to both increase and decrease the fluxes of atmospheric gases and





particles to and from the ocean, along with its high variability both seasonally and spatially, means that accurate determination of the presence of the SML and how SA relates to these processes is vitally important.

In this study, we first assess SA measurements obtained from voltammetry on a variety of (and mixtures of) model surfactants, including TX-100. We evaluate how reliably SA can be used to determine surfactants at seawater concentrations, how the
technique responds to different surfactants, and how suitable TX-100 is as a reference surfactant. We then determine surface tension and SA for natural water samples in both bulk water and the SML, and compare the surfactant levels determined by these two techniques.

## 2. Experimental

### 2.1 Sampling seawater and lake water

Sampling was performed in a variety of locations in the UK; at the University of York lake on the Heslington West campus between January and March 2018, from the North Sea at 6-13 km off Flamborough Head, Yorkshire between April 2018 and February 2019, and from the English Channel, approximately 2 km east of Penlee Point, near Plymouth, in April-May 2018.

All SML samples were collected with a Garrett screen (16 mesh, 0.80 x 0.54 m). These were initially dipped 3-5 times in the
seawater to rinse them and the water collected from these dips was used to rinse Winchester collection flasks. The screen was submerged and withdrawn from the water as parallel to the surface as possible. Excess water was allowed to drain from the screen until only the film within the wire mesh remained, which was emptied into the Winchester collection flasks. The thickness of the SML samples, as calculated from the volume collected per dip, ranged between 155 and 225 µm, which is typical for Garrett screen sampling. Sub-surface water (SSW) samples were collected from the North Sea using a HPDE brown
bottle attached to an extendable pole off the side of the boat on the windward side to reduce the impact of pollution from the boat on the samples. These samples were then poured into a 2.5 L Winchester flask, rinsed 3 times with the sample water before filling. The SSW samples collected from the English Channel were collected using a pump, with an inlet located 5 m below the surface.

All equipment was soaked in an acid bath and washed with Milli-Q MQ (18.2 MΩ cm) water before use. All of the SML and SSW samples collected for SA analyses, and most of the samples for tensiometry analyses, were filtered using GFF combusted filters with a pore size of 0.7 µm, then stored in a freezer at -20 ºC for later measurements. The surface tension measurements of the unfiltered samples were performed as soon as possible, within 24 hours of collection. Note that filtering removes particulate surfactants (Ćosović and Vojvodić, 1987; Wurl et al., 2011; Pereira et al., 2016) and thus changes the surface
tension and surfactant activity. Thus, only unfiltered samples reported here can be used as a true reflection of surfactant activity

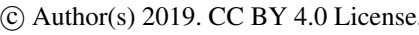

in natural waters. However, since the filtering process was the same for both the surface tension and voltammetry measurements, we use the filtered samples for comparison and evaluation of these techniques.

## 2.2 Surfactant activity (SA) by voltammetry

Surfactant activity values were measured by voltammetry using a Metrohm 663 VA Stand paired with an IME623 µAutolab

Type III Potentiostat/Galvanostat. The voltammeter consists of an Ag/AgCl reference electrode containing 3M potassium chloride solution, platinum auxiliary electrode and a hanging mercury drop electrode (HMDE). The voltammeter is connected to a nitrogen gas cylinder which is used to purge samples and to provide pressure for formation of mercury drops from the capillary. Before each run, the Teflon cell was washed with hot tap water then deionised water and submerged in a 4% HCl bath for 5 minutes. The cell was then rinsed with MQ water, dried and then charged with a blank solution of sodium chloride

(3M, 1.83 mL), sodium hydrogen carbonate (10 mM, 2.00 mL) and MQ water (6.17 mL). The SA for the samples was determined by measuring the decrease in capacitance current ($\Delta I$) at -0.6 V caused by additions of TX-100. The volumes of sample and TX-100 standard added varied each time to achieve capacitance current values within the measurable range of the instrument. After the addition of each new component to the vessel, the solution was stirred for 5 minutes to homogenise it before any measurements were made. Table 1 shows the typical parameters for the voltammetry runs.

**Insert Table 1 here**

## 2.3 Surface Tension Measurements

Surface tension was measured using Du Noüy Ring tensiometry (Harkins et al., 1917; Noüy, 1925) performed using an Attension Sigma 70 instrument with thermocouple (density resolution: $1 \times 10^{-4}$ g cm$^{-3}$, force resolution: 0.1 µN) which was calibrated daily. The short-term precision of the instrument, as measured from ~100 repeat measurements, was typically 0.05-

0.1 mN m$^{-1}$.

After the instrument had been calibrated with the calibration weight, the shallow sample vessel (~30 mL) was washed with acetone or ethanol, then submerged in a 4 % HCl acid bath before being rinsed with MQ (18.2 MΩ cm) water and dried in an oven at 120 ºC. The vessel was then cooled under a flow of nitrogen and placed in the instrument. The sample to be tested was

then poured into the vessel until it was full (~30 mL). The thermocouple was then inserted into the sample and the sample was left for at least 5 mins to equilibrate. After the sample had equilibrated the platinum ring was dipped and shaken in MQ (18.2 MΩ cm) water and burned in a Bunsen flame before being placed in the machine and the run initiated, using the parameters shown in Table 2.

**Insert Table 2 here**

### 2.4 Tensiometry calibration

The accuracy of the tensiometer was tested by measuring the surface tension of several reference compounds and comparing the results to values reported in literature, shown in Table 3 and Fig. 2. The results show that the surface tensions measured are slightly lower than reported values, with the deviation being greater at higher surface tensions. The linear fit in Fig. 3 has a gradient of 1.0347, which is used as a correction factor for the surface tension measurements of seawater samples discussed later.

**Insert Table 3 here**

**Insert Figure 2 here**

Tensiometry measurements of TX-100 over the concentration range $0 - 200$ mg $L^{-1}$ were performed to determine the experimental critical micelle concentration (CMC) and compare it to literature values (Fig. 3). The CMC represents surface saturation of the compound and so can be determined as the concentration after which an increase in concentration does not decrease the surface tension and micelles start to form from the surface excess concentration.

**Insert Figure 3 here**

The CMC of TX-100 in MQ water was determined as approximately 180 mg $L^{-1}$ or 0.288 mM. This is similar to values found in other studies using alternate methods, such as 0.30 mM determined using cyclic voltammetry (Mandal et al., 1988) and 0.27 mM determined using spectrometry (Karimi et al., 2015). Note that CMCs are much lower in salt water than in pure water because the surfactant molecules are less soluble in the greater ionic strength salt solution and so more readily adsorb at the air-water interface to hide their hydrophobic tails. This means that at a given surfactant concentration, there are a greater number of surfactant molecules at the surface in the salt water solution than in pure water, resulting in a lower surface tension and a lower bulk concentration needed to reach surface saturation.

### 3. Results

#### 3.1 SA Voltammetry: Linear range, effect of deposition time, and response to different surfactants

A potential drawback of the SA technique is that the adsorption effect at the electrode becomes saturated at relatively low surfactant concentrations. Thus, it is important to select measurement conditions that correspond to the rising part of the calibration curve and below the surface saturation level (Ćosović and Vojvodić 1982, 1998).

#### 3.1.1 Effect of deposition time

Voltammetry calibration curves of TX-100 using deposition times ranging from 15 s to 120 s are shown in Fig. 4. Each of the ΔI – concentration curves show a linear increase in ΔI with increasing concentration until approximately 400 nA at which point the curve begins to flatten due to the electrode becoming saturated. The ΔI – concentration slope differs with the chosen deposition period, becoming less readily saturated at lower deposition times. The longer deposition times result in higher





suppression of capacitance current at a given concentration because the surfactant molecules have more time to diffuse through the solution and adsorb onto the electrode surface, resulting in greater surface coverage.

A slight deviation from linearity is also seen at the lowest concentrations, which was also observed by Ćosović and Vojvodić (1982) and is clearer in their calibration curves due to the presence of more points at these lower concentrations. Overall, it is

clear that even at deposition times of 15 s, saturation of TX-100 occurs at ~ 2 mg L$^{-1}$, which is at the higher end of observed SA values in the SML (Wurl et al., 2011; Sabbaghzadeh et al., 2017). The response to surfactants can also be non-linear before this saturation level.

**Insert Figure 4 here.**

### 3.1.2 SA response to different surfactants

Voltammetry calibration curves for a variety of surfactants using a 15 s deposition time are shown in Fig. 5. The curves all have the same general shape, showing a linear or near-linear increase in ΔI with increasing concentration up until a saturation point. However, they each have different gradients and saturation concentrations, showing that the adsorption behaviour of the different surfactants is not identical. Cetyltrimethylammonium chloride (CTAC) has higher efficiency than TX-100 and linoleic acid higher than both. The higher the surface adsorption efficiency of the surfactant/mixture, the lower the effective

linear range of the SA technique: for linoleic acid, saturation of the electrode occurs at < 1 mg L$^{-1}$.

The behaviour shown in Fig. 5 differs from a similar plot by Ćosović and Vojvodić (1982) in which some surfactants only begin to show an increase in ΔI above the saturation concentration of TX-100. This may be related to the type of surfactants used; the ones studied by Ćosović and Vojvodić (1982) were all complex biological molecules containing more hydrophilic

groups than TX-100, while the surfactants used in this study are all lipid-like linear hydrophobic structures with a hydrophilic head and so more surface active.

**Insert Figure 5 here**

The gradients of the rising slopes of the voltammetry calibration curves give the decrease in capacitance current, and therefore HDME surface coverage, per unit concentration and so indicate the surface adsorption efficiency of the surfactant/mixture.

Figure 6a shows that the technique is generally more responsive to low molecular weight than high molecular weight surfactants, as noted previously (Ćosović and Vojvodić, 1982). Since the efficiency in Fig. 6a is defined in terms of the mass concentrations, this trend may arise due to the higher molecular weight surfactants having a lower number of individual molecules per unit mass, resulting in fewer molecules in solution able to adsorb onto the surface. If the decrease of capacitance current were entirely dependent on the number of molecules in solution, the efficiency defined in terms of the molar

concentration would be expected to be equal for all the different surfactants, since at any given molar concentration there would be the same number of molecules present in solution. Figure 6b shows that this is not the case and that the higher molecular weight (MW) surfactants have higher efficiency in terms of the molar concentration. This may be because the higher





MW molecules are generally larger and so occupy more space at the electrode surface, resulting in each individual molecule covering more of the surface. The extra mass of these larger molecules mostly comes from longer carbon chains and therefore

molecular weight could be considered as a loose representation of the surfactant's hydrophobicity. This supports the trend seen in Fig. 6b since the more hydrophobic molecules will more readily adsorb onto the electrode surface, resulting in a higher efficiency. Considering Fig. 6a and 6b together, the size and hydrophobicity of each individual surfactant molecule as well as the total number of molecules in solution appear to each have an impact on the efficiency of detection by this technique.

**Insert Figure 6 here**

**3.1.3 Surfactant mixtures**

Voltammetry calibration curves for solutions containing multiple surfactants are shown in Fig. 7. The efficiencies of the mixtures of TX-100 with CTAC and with linoleic acid are intermediates of those for the pure surfactants, indicating a contribution from each surfactant towards the electrode coverage. This indicates that the presence of TX-100 does not interfere with the adsorption of the second surfactant onto the HDME and that the relative adsorption of each surfactant onto the

electrode is related to its relative concentration. However, this contribution is not linear: the 50:50 mixtures both have efficiencies closer to that of TX-100 than to the other surfactant. This may indicate that in these cases TX-100 is contributing to a higher proportion of the electrode coverage than would be expected based upon its relative concentration.

**Insert Figure 7 here**

**3.2 Surface tension and surface activity (SA) measurements of lake water and seawater**

Table 4 shows the surface tension, surface film pressures ($\Delta\gamma$), and surfactant activity results for the seawater and lake water samples, along with enrichment factors (EF), calculated as the ratio of the SA in the SML to the SSW. For seawater, $\Delta\gamma$ was calculated as the difference in surface tension between the sample and $\gamma_0$, i.e. $\gamma$ of saltwater without any surfactants, calculated at the appropriate laboratory temperature and with a salinity of 69.5 g kg$^{-1}$ (Nayar et al., 2014). For the lake water samples, the reference surface tension $\gamma_0$ was the value for pure water at 15$^\mathrm{o}$ C of 73.49 mN m$^{-1}$ (Vargaftik et al., 1983).


The SML samples all have lower surface tension and higher SA values than their respective SSW samples, indicating an enrichment of surfactants in the SML, as expected. The surface tensions of all the SML samples increase upon filtration. Determination of the surface film pressure before and after filtering indicates that filtering removed ~ 20-75% of the surface-active material from the SML.

Only three unfiltered measurements of surface tension and surface film pressure in the ocean SML (coastal North Sea) were made. The surface tensions in the SML were 53.3, 58.1 and 68.1 mN m$^{-1}$, corresponding to surface film pressures ($\Delta\gamma$) of 22.3, 16.9 and 6.8 mN m$^{-1}$. The two highest values of $\Delta\gamma$ were determined in April and May and the lower value in October, possibly indicating greater biological activity in spring producing surface active material. These $\Delta\gamma$ values are reasonably



comparable to the maximum values observed by Schmidt and Schneider (2011) in coastal waters of the southern Baltic Sea of
up to 10 mN m$^{-1}$.

The Surfactant Activity values determined here, of filtered samples, cannot be directly compared with unfiltered SA
measurements of seawater for the reasons explained above.  Nevertheless, they are within the range of previous observations
(SSW mean and 1 s.d. of 0.35$\pm$0.11 mg L$^{-1}$ TX-100 eq. and SML mean and 1 s.d. 1.21$\pm$0.91 of mg L$^{-1}$ TX-100 eq.) indicating
that the SA of these samples are representative of ocean sub-surface and SML waters (see sect. 1.2).
**Insert Table 4 here**

The relationship between surface film pressure and surfactant activity in the SML (SA$_{SML}$) is shown in Fig. 8. This shows that
while an increase in surface film pressure is generally accompanied by an increase in SA, the SA values plateau above
concentrations of ~2 mg L$^{-1}$ TX-100 eq., as the surface film pressures continue to increase. This is consistent with our results
(as shown in Fig. 4, 5 and 7) that the SA technique saturates above ~1-2 mg L$^{-1}$ TX-100 eq. at 15 s deposition time, with the
exact saturation point depending on the surfactants present.

Figure 9 shows that there is also no clear relationship between the enrichment factors (EF$_{SA}$) calculated from the SA and the
surface film pressure. Indeed, the EF$_{SA}$ values show limited variability across quite a wide range of surface film pressures
between 0 - 10 mN m$^{-1}$. The lack of relationship is not surprising, given that EF is a measure of enrichment rather than that of
surfactant concentration. However, these data caution against the use of EF as an indicator of the ability of the SML to impact
on gas-exchange or on any absolute property of the SML.
**Insert Figure 8 here**

**Insert Figure 9 here**

**4. Conclusions**

The surfactant-enriched layer on the surface of the ocean is a largely uncharacterised system that is thought to have a large
impact upon many processes important to both human health and the planetary climate. Currently, the only measurements of
these surfactants at the ocean basin scale are based on surfactant activity (SA) from alternating current voltammetry. Here we
have attempted to better characterise such measurements and put them in the context of surface film pressure.

The response of SA voltammetry to surfactants becomes saturated depending upon the deposition rate chosen for the electrode.
Shorter deposition times result in lower sensitivity but higher saturation concentrations, corresponding to a higher linear range.
However even at the lowest deposition time used in this study of 15 s, which is typically used in SA measurements of the SML
(Sabbaghzadeh et al., 2017; Wurl et al., 2011),  saturation occurs at surfactant concentrations of around 1- 2 mg L$^{-1}$, which are
lower than those observed in highly enriched SML samples.

The technique becomes more readily saturated by lower molecular weight surfactants, such as C18 fatty acids, than higher molecular weight surfactants such as TX-100, when considering their mass concentration, because the technique is generally more responsive to low molecular weight surfactants. Our results suggest that the response of the HDME, which is related to

its coverage during deposition, is simultaneously dependant on the number of surfactant molecules in solution and their size. The use of any single surfactant, such as TX-100, as an external calibrant for SA voltammetry does not give accurate concentration values for other surfactants. Therefore, SA values should not be considered as a direct reflection of surfactant concentrations, and may not be comparable between samples obtained from different locations, which may contain a mixture of different surfactants.


Surfactant activity determined by voltammetry has only a weak relationship to surface film pressure, showing that it does not linearly respond to the surface activity of a sample, but rather to compounds that adsorb most readily onto the mercury electrode, some of which may not as readily adsorb at the air-water interface. The fact that the SA response becomes saturated at enriched SML surfactant concentrations is also likely to mask any relationship between SA and surface tension or surface

film pressure. Further, the enrichment (EF) of surfactants of the SML compared to underlying water as determined by SA measurements shows no relationship with surface film pressure over a range typically observed in ocean waters, thus cautioning against the use of EF as an indicator of the ability of the SML to impact on gas-exchange or on any absolute property of the SML.

Currently, despite its limitations, SA by voltammetry is the only practical method available for determination of surfactants at sea. Investigation of further methods should be a matter of priority for the sea-air exchange community.

**Acknowledgements**

We acknowledge the UK Natural Environment Research Council (NE/N009983/1) for funding. Thanks to Adam Saint, David Loades, Timothy Pilbeam and Dr Rosie Chance at the University of York, for their help in collecting seawater and lake water

samples, and to Dr Stuart Young at the University of York for the design and construction of the subsurface sampler. We gratefully acknowledge Daniel Philips and Dr Frances Hopkins from the Plymouth Marine Laboratory for the collection of the English Channel samples.

**Data availability**

Data sets are available upon request by contacting the corresponding author.



## Author contribution

The work was carried out by LK and IR as undergraduate project students under the supervision of LT and LC. The paper was written by LK and LC with input from the other authors.

## Competing Interests

The authors declare that they have no conflict of interest.

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



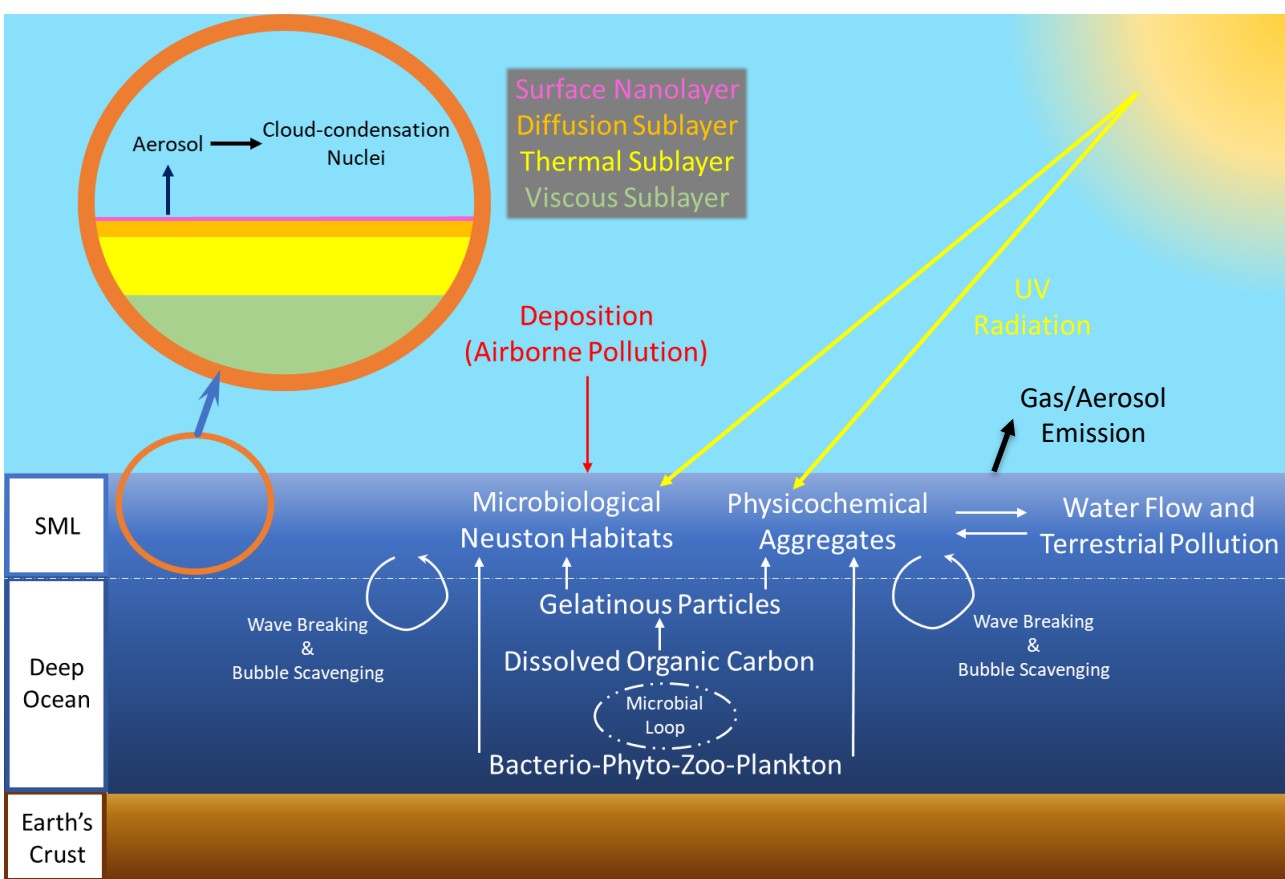

**Figure 1: An illustration of the layering and general processes occurring in the SML and its interactions with surrounding systems.**
**Based on Soloviev and Lucas (2014) and Cunliffe et. al. (2013).**



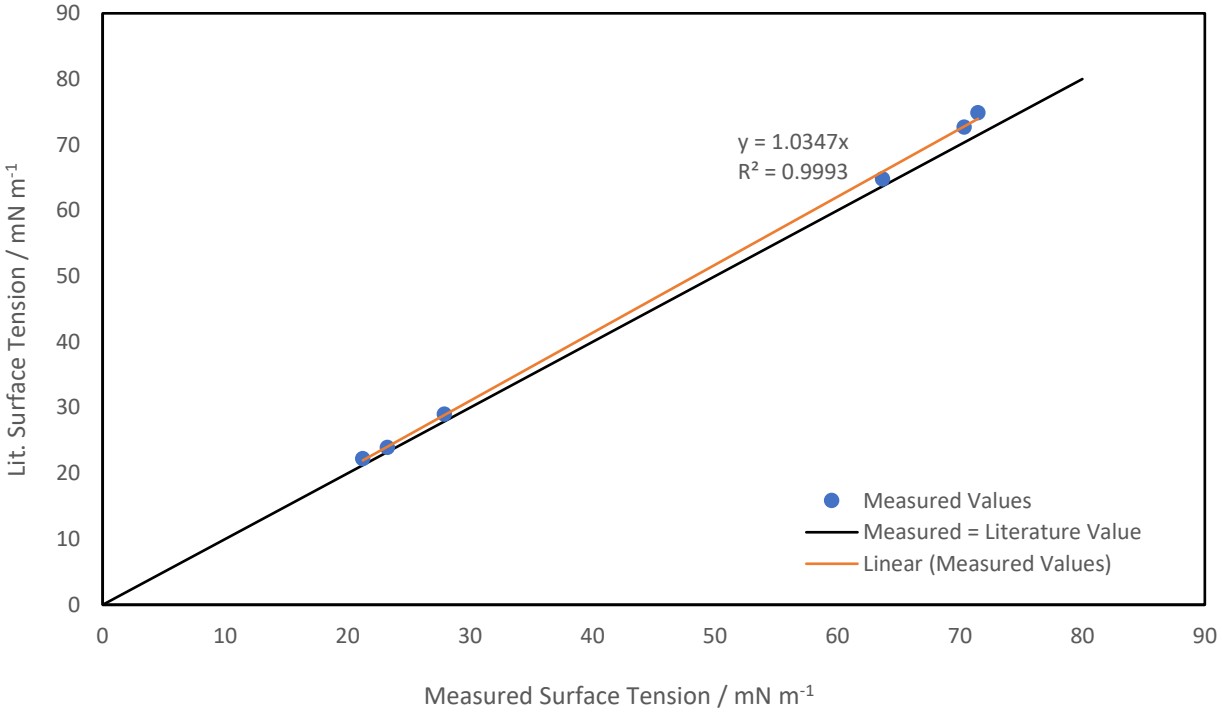

**Figure 2. Comparison between the surface tensions of reference compounds measured by the Attension Sigma 70 instrument used in this study and those reported in literature (values and standard deviation are shown in Table 3).**

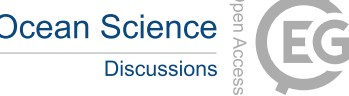

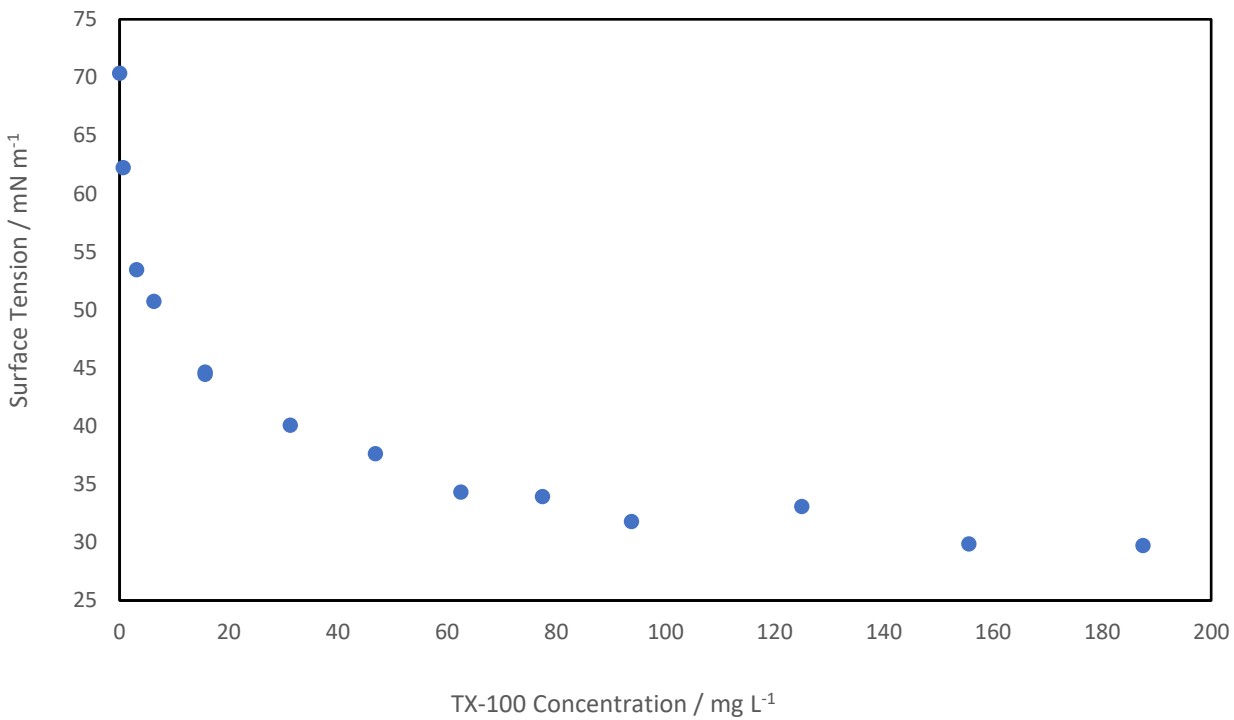

**Figure 3. Surface tension of a calibration series of TX-100 in MQ water.**





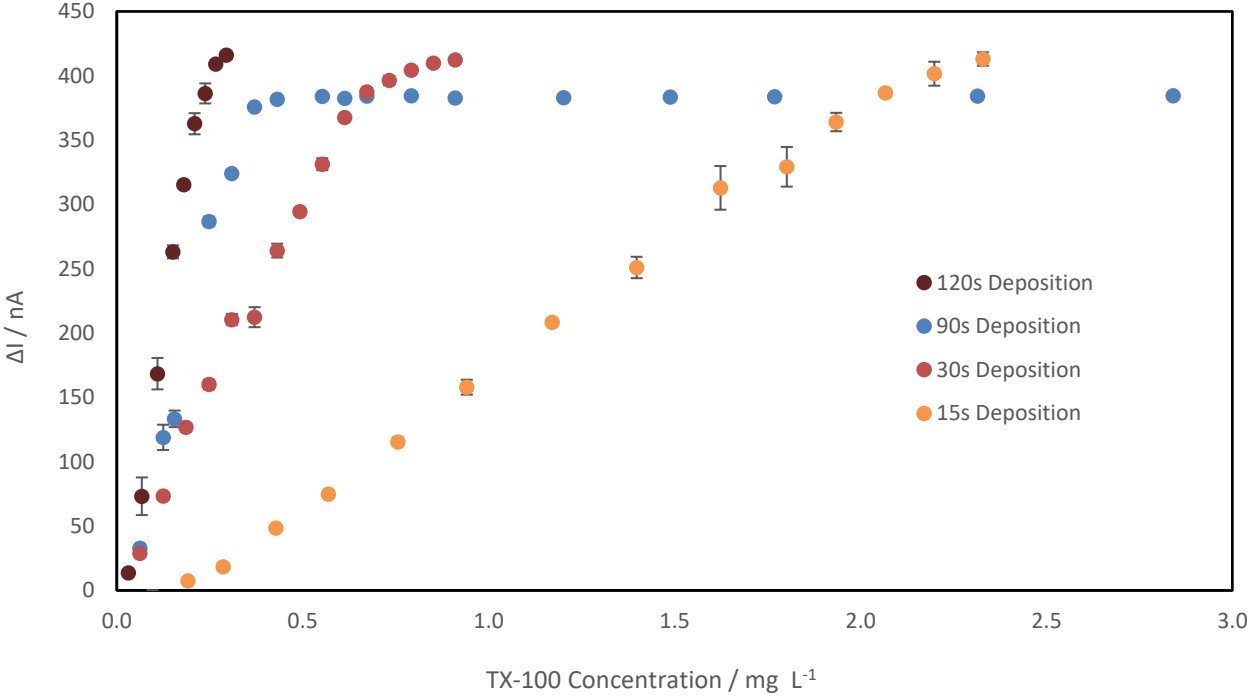

**Figure 4. Example voltammetry calibration plots of TX-100 using different deposition times. Error bars represent 1 standard deviation of 4 repeat measurements.**



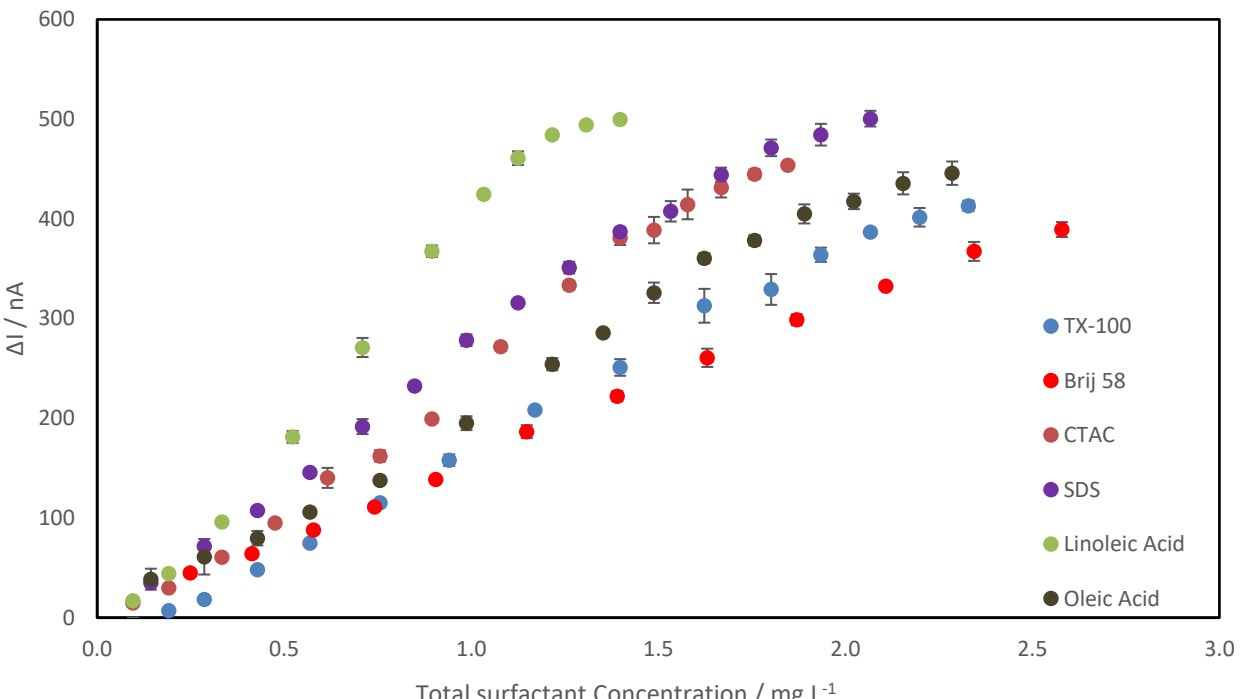

**Figure 5. Voltammetry calibration plots of different surfactants using 15 s deposition time. Brij 58 = polyethylene glycol hexadecyl ether, CTAC = cetyltrimethylammonium chloride, SDS = sodium dodecyl sulfate. Error bars represent 1 standard deviation of 4 repeat measurements.**






**Figure 6. Relationship between the average molecular weight of surfactants in solution and the gradient (*m*) of the voltammetry calibration plot in terms of (a) mass concentration (b) molar concentration. For surfactant mixtures, the MW is calculated as the total surfactant mass concentration divided by the total surfactant molar concentration (mg L⁻¹/mM).**



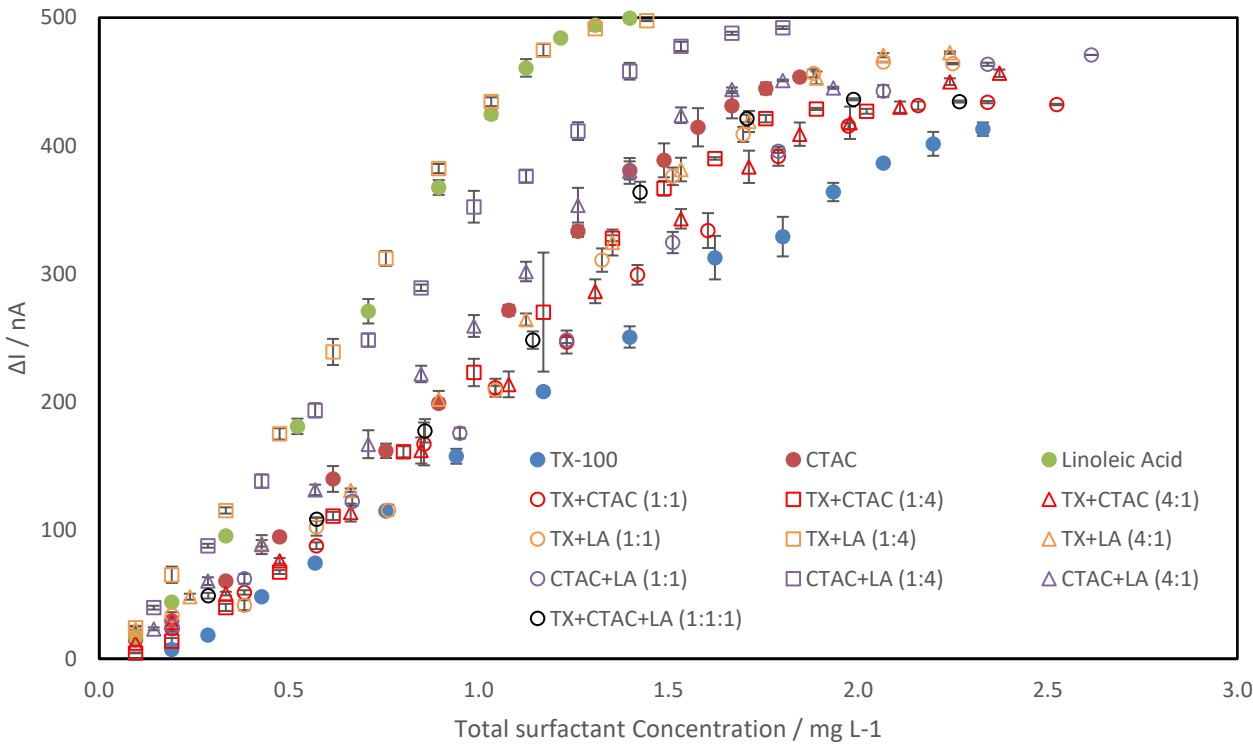

**Figure 7. Voltammetry calibration plots of surfactants and their mixtures in terms of total surfactant concentration. The numbers in brackets indicate the respective mass ratios of each surfactant. TX = TX-100, LA = Linoleic Acid, CTAC = cetyltrimethylammonium chloride. Error bars represent 1 standard deviation of 4 repeat measurements.**



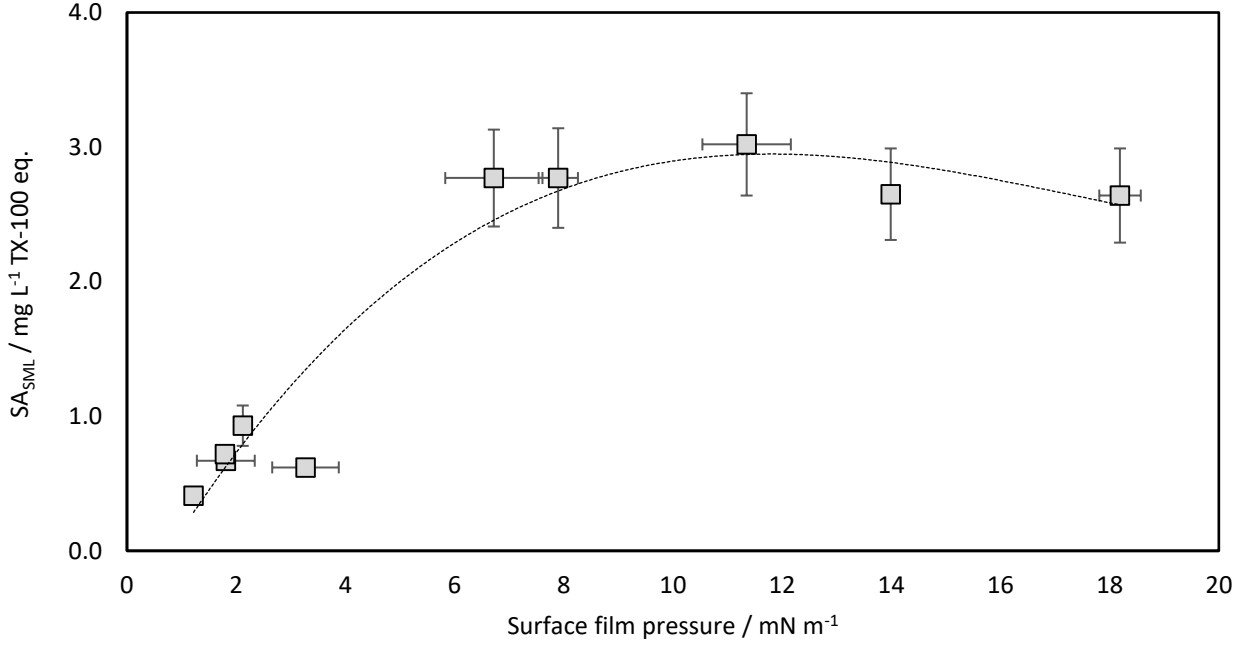

**Figure 8. Relationship between surface film pressure (Δγ) and surfactant activity (SA) determined by external calibration using TX-100 with a 15 s deposition time. The dotted line is a polynomial fit to the data.**





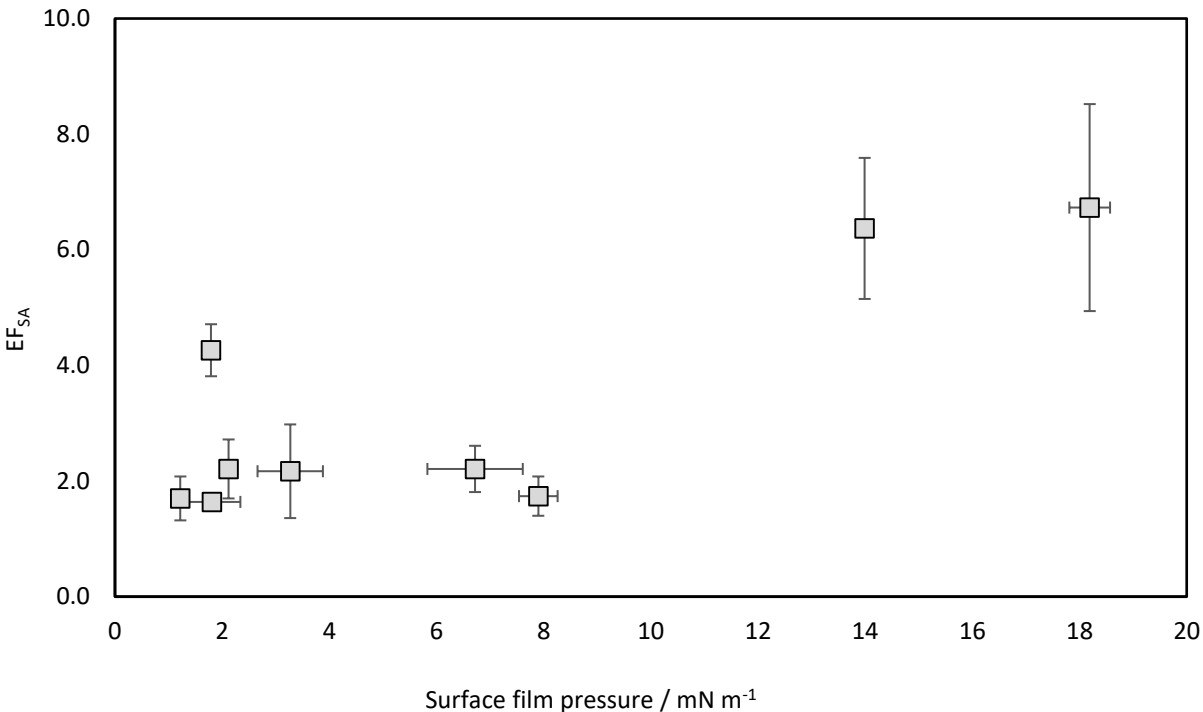

**Figure 9. Relationship between surface film pressure (Δγ) and surfactant enrichment in the SML calculated from surfactant activity (EF_SA).**






**Table 1. Parameters for a typical voltammetry run.**

| Parameter | Value |
| --- | --- |
| Start potential | -0.6 V |
| Stop potential | -0.625 V |
| Step | -0.002 V |
| Modulation amplitude | 0.012 $V_{RMS}$ |
| Modulation time | 0.15 s |
| Frequency | 70 Hz |
| Interval time | 0.55 s |
| Scan rate | 0.0036 V s$^{-1}$ |
| Stirrer speed | 3 |
| Drop size | 3 |
| Repeats per conc. | 4 |









**Table 2. Parameters used for Du Noüy Ring surface tension measurement.**

| Parameter | Value |
| --- | --- |
| Speed Up | 7.5 mm min$^{-1}$ |
| Speed Down | 20.0 mm min$^{-1}$ |
| Dwell Down | 5.0 % |
| Points | 10 |
| Detection Range | 2.0 mN m$^{-1}$ |










**Table 3. Surface tensions of reference compounds measured by the Attension Sigma 70 instrument at 20 $\pm$1 ºC.**

| Sample | Surface Tension / mN m$^{-1}$ | Std. Dev. / mN m$^{-1}$ | Lit. Surface Tension at 20ºC / mN m$^{-1}$ |
|---|---|---|---|
| Ethanol | 21.240 | 0.047 | 22.27 (Richards and Coombs, 1915) |
| Ethyl Acetate | 23.246 | 0.069 | 23.95 (Mumford and Phillips, 1950) |
| Acetonitrile | 27.888 | 0.064 | 29.04 (Harkins et al., 1917) |
| Glycerol | 63.686 | 0.057 | 64.800 (Vargo et al., 1991) |
| Pure Water | 70.335 | 0.089 | 72.703 (Vargaftik et al., 1983) |
| Salt Water | 71.478 | 0.092 | 74.876 (Nayar et al., 2014) |









**Table 4. Surface tension, surfactant activity, surface film pressure and EF$_{SA}$ values for lake and seawater samples collected in this study. Error bars represent 1 standard deviation of the data.**

| Date | Sample | Surface Tension / mN m⁻¹ | | Surfactant Activity / mg L⁻¹ TX-100 eq. | Surface Film Pressure / mN m⁻¹ | | EF$_{SA}$ |
| | | Unfiltered | Filtered | Filtered | Unfiltered | Filtered | Filtered |
|---|---|---|---|---|---|---|---|
| **University of York lake water (53°95N, 1°05W)** | | | | | | | |
| 29.01.18 | SML | 33.66±0.01 | 62.14±0.81 | 3.02±0.38 | 39.83±0.16 | 11.35±0.81 | - |
| | SSW | 73.23±0.16 | 73.55±0.04 | - | | | |
| 01.02.18 | SML | 43.48±0.10 | 66.77±0.88 | 2.77±0.36 | 30.01±0.10 | 6.72±0.89 | 2.21±0.40 |
| | SSW | 73.09±0.02 | 72.77±0.14 | 1.25±0.16 | | | |
| 29.03.18 | SML | 56.83±0.49 | 65.59±0.13 | 2.77±0.37 | 16.66±0.60 | 7.90±0.36 | 1.74±0.34 |
| | SSW | 73.32±0.34 | 73.15±0.34 | 1.59±0.22 | | | |
| **Coastal North Sea (53°95N, 0°02W)** | | | | | | | |
| 24.04.18 | SML | 52.56±1.78 | 56.68±0.32 | 2.64±0.35 | 22.32±1.78 | 18.19±0.38 | 6.73±1.79 |
| | SSW | 73.58±0.10 | 73.84±0.21 | 0.39±0.09 | | | |
| 04.05.18 | SML | 58.11±0.22 | 60.89±0.01 | 2.65±0.34 | 16.77±0.32 | 13.99±0.04 | 6.37±1.22 |
| | SSW | 74.22±0.23 | 73.15±0.04 | 0.42±0.06 | | | |
| 15.08.18 | SML | - | - | 1.07±0.17 | - | - | 2.18±0.50 |
| | SSW | - | - | 0.49±0.08 | | | |
| 17.10.18 | SML | 68.08±0.21 | 72.76±0.12 | 0.93±0.15 | 6.80±0.22 | 2.12±0.16 | 2.21±0.51 |
| | SSW | 74.27±0.05 | 74.76±0.10 | 0.42±0.07 | | | |
| 27.02.19 | SML | - | 73.66±0.06 | 0.41±0.06 | - | 1.22±0.08 | 1.70±0.38 |
| | SSW | - | 74.82±0.05 | 0.24±0.04 | | | |
| **Coastal English Channel (50°19N, 4°10W)** | | | | | | | |
| 05/18 | SML | - | 73.08±0.41 | 0.72±0.05 | - | 1.79±0.43 | 4.26±0.45 |
| | SSW | - | 73.77±0.10 | 0.17±0.01 | | | |
| 08.05.18 | SML | - | 71.61±0.61 | 0.62±0.03 | - | 3.27±0.61 | 2.17±0.81 |
| | SSW | - | 73.86±0.05 | 0.29±0.11 | | | |
| 21.05.18 | SML | - | 73.06±0.42 | 0.67±0.04 | - | 1.81±0.53 | 1.64±0.15 |
| | SSW | - | 74.08±0.32 | 0.41±0.03 | | | |