# Peer review of "The Determination of Surfactants at the Sea Surface"

_Ocean Science, 2019_

## Referee Comment (RC1) · Anonymous Referee #1 · 20 Aug 2019

Review of ms. The Determination of Surfactants at the Sea Surface by Leon King, leuan J. Roberts, Liselotte Tinel and Lucy J. Carpenter. King and coauthors measured surfactants in oceanic samples by ac voltammetry and by surface film pressure and done experiments with model SAS. I am sorry to tell that paper should be rejected due to few reasons: I learned nothing, majority of model experiments is already published in 1980ies. I am not sure if this paper intended to be methodological (if yes than nothing new came up) or ecological (too few data). Specific comments: Abstract Majority of the abstract is written as Introduction. L 16 - Method is calibrated and not SA 1 Introduction Introduction is too long. Huge part of it is book knowledge. Line 26/27 – I suggest replacing oceanic mixed layer with sea/ocean. Lines 26 and 28 – If it is stated that SML comprises the top 10-1000  $\mu$ m than the viscous sublayer (>1000  $\mu$ m) would not be within SML

2. Experimental Paragraph on Reagents should be added lines 174 - 179 - What was

the volume of the SML sampled? lines 195 – 196 – The voltammeter does not consist of the electrodes. I suggest replacing this sentence with: The experiments were performed in a three-electrode system with an Ag/AgCl reference electrode containing 3M potassium chloride solution, platinum auxiliary electrode and a hanging mercury drop electrode (HMDE). Lines 196 – 197 – The voltammeter is NOT connected to a nitrogen gas cylinder. Electrochemical cell is connected to a nitrogen gas cylinder. The only one reason for the using nitrogen gas is to provide pressure for formation of mercury drop. I suggest removing this sentence. Lines 200 – 204 – The method for the measurement of SA is not described well and should be improved. If I understand properly they used method of the standard addition. As I am aware, this method is used by the O. Wurl group, and should be cited. However, this method is much more demanding than those published by Cosovic. Line 203 - vessel is not proper electrochemical word. It should be "the electrochemical cell" or "the cell" L 213 – to remove: (18.2 M $\Omega$  cm) L 215 – to remove: ( $\hat{a}$ Lij30 mL) L 216 – to remove: (18.2 M $\Omega$  cm) L 220 – 239 – I really do not understand why the authors were interested in the determinination of the TX-100 CMC? The CMC of TX100 is two order of magnitude higher that that one found in the real samples. 3. Results L 301 – EF is already explained at line 105 L 302 –  $\gamma$ 0 is already explained at line 72 L 310 - ???Only three unfiltered measurements. It is not unfiltered measurements but rather measurements of unfiltered samples

4. Conclusions L 340 - there is no method called SA voltammetry

---

## Referee Comment (RC2) · Matthew Salter (Referee) · 6 Sep 2019

**Summary**

King *et al.* present surfactant activity (SA) and surface tension measurements of a series of sea surface microlayer (SML) and corresponding underlying water (ULW) samples collected from a lake, the coastal North Sea and the coastal English channel along with a series of model surfactants at varying concentrations. As described by the authors, the purpose of these measurements was to ascertain the suitability of SA measurements for the determination of the total concentration of surfactants present given that this technique has become more widespread in recent years.

A key finding of the study is the non-linear relationship between surface film coverage (also referred to as surface film pressure) and the SA of the samples. This suggests

that the two parameters are responding differently to the amount or types of surfactants present in the samples.

Unfortunately, it is my view that the manuscript suffers from serious deficiencies and I can only recommend that the manuscript be rejected. In the following, I describe the major issues I have identified along with some more minor points.

**Major comments**

**Major gaps in the literature review -** To state that there have "been only three major studies of the surface tension of seawater in the last 100 years" is quite simply incorrect. During a 10-minute literature search I was able to find numerous studies that have presented surface tension measurements of seawater (e.g. Sturdy and Fischer, 1966; Barger *et al.*, 1974; Hühnerfuss *et al.*, 1977; Zhengbin *et al.*, 1998; Wei and Wu, 1992).

**Flaws in the key findings of the paper -** As stated in the introduction to this review, a key finding the authors make is presented in Fig. 8 - namely that the measured surface tension of the samples are not linearly correlated with the SA measurements. As stated by the authors, depending on the concentration and surface activity of the surfactants present in a sample, along with the chosen deposition time, the mercury drop can become saturated. As such, a range of deposition times should be used along with dilution of samples if required to bring the samples into the "linear range" of the instrument. The opposite issue can arise when measurements of surface tension are made - this technique is relatively insensitive when the concentration or surface activity of surfactants is low (e.g. see the literature cited above). Given that a number of the SA measurements made by the authors were clearly at conditions where the mercury drop was saturated while they were in the ideal range for surface tension measurements means that direct comparison is impossible and the conclusion that the two techniques are responding differently to the amount or type of surfactants present cannot be made.

With regards the low sensitivity of surface tension measurements, it is worth pointing out that there are more accurate indicators of the relative amount of surfactants actually absorbed at the air-water interface than standard surface tension measurements when relatively soluble, highly oxygenated surfactants are present (i.e. directly relevant for SML and gas exchange/aerosol studies). An example of such an approach are measurements of the surface pressure following compression using Langmuir-Blodgett troughs (see e.g. Goldman *et al.* (1988); Frew *et al.* (1990). It would be interesting to compare this parameter to appropriate measurements of SA and the authors may be interested to note that something similar has been presented by Kozarac *et al.* (2003).

**Lack of novelty -** Aside from the flawed comparison of surface tension to SA the remainder of the paper can only be described as a repetition of previous work. The effect of deposition time on SA along with the response of SA to different model surfactants was presented by Cosović and Vojvodić (1982) and others since. As discussed above, the surface tension and SA measurements of lake and seawater have been made on numerous occasions and the limited number presented here without other contextual data do not add to our understanding greatly.

**Inappropriate article title -** The title of the manuscript is inappropriate. Upon first reading, I assumed that the manuscript contained a review of the literature on the topic. Further, I think a reader that is not particularly familiar with the topic may assume that the manuscript presents a new method for "the determination of surfactants at the sea surface". As such, the title must be amended to something more appropriate.

**Other minor points**

**P5, line 149 -** The authors should be aware that Wilson *et al.* (2015) did not determine how effective marine aerosol are as ice nuclei they determined how effective SML samples were at nucleating ice.

**P2, line 33 -** Throughout the manuscript the authors refer to the same parameter as both "surface film coverage" and "surface film pressure". Although both terms are used in the literature I would urge consistency in the manuscript.

**References**

Barger, W. R., Daniel, W. H. *et al.*, 1974, *Deep-Sea Research and Oceanographic Abstracts*, 21, 83–89, doi:10.1016/0011-7471(74)90022-9.

Cosović, B. and Vojvodić, V., 1982, *Limnology and Oceanography*, 27, 361–369, doi:10.4319/lo.1982.27.2.0361.

Frew, N. M., Goldman, J. C. *et al.*, 1990, *Journal of Geophysical Research*, 95, 3337, doi:10.1029/jc095ic03p03337.

Goldman, J. C., Dennett, M. R. *et al.*, 1988, *Deep Sea Research Part A, Oceanographic Research Papers*, 35, 1953–1970, doi:10.1016/0198-0149(88)90119-7.

Hühnerfuss, H., Walter, W. *et al.*, 1977, *Journal of Physical Oceanography*, 7, 567–571, doi:10.1175/1520-0485(1977)007<0567:otvost>2.0.co;2.

Kozarac, Z., Ćosović, B. *et al.*, 2003, *Colloids and Surfaces A: Physicochemical and Engineering Aspects*, 219, 173–186, doi:10.1016/S0927-7757(03)00032-3.

Sturdy, G. and Fischer, W. H., 1966, *Nature*, 211, 951–952, doi:10.1038/211951b0.

Wei, Y. and Wu, J., 1992, *Journal of Geophysical Research*, 97, 5307, doi:10.1029/91jc02820.

Wilson, T. W., Ladino, L. A. *et al.*, 2015, *Nature*, 525, 234–238, doi:10.1038/nature14986.

Zhengbin, Z., Liansheng, L. *et al.*, 1998, *Journal of Colloid and Interface Science*, doi:10.1006/jcis.1998.5538.